# A LAT-Based Signaling Complex in the Immunological Synapse as Determined with Live Cell Imaging Is Less Stable in T Cells with Regulatory Capability

**DOI:** 10.3390/cells10020418

**Published:** 2021-02-17

**Authors:** Yikui Li, Helen M. Tunbridge, Graham J. Britton, Elaine V. Hill, Parisa Sinai, Silvia Cirillo, Clare Thompson, Farnaz Fallah-Arani, Simon J. Dovedi, David C. Wraith, Christoph Wülfing

**Affiliations:** 1School of Cellular and Molecular Medicine, University of Bristol, Bristol, BS8 1TD, UK; Yikui.Li@bristol.ac.uk (Y.L.); Helen.tunbridge@gmail.com (H.M.T.); graham.britton@mssm.edu (G.J.B.); elaineavramenko@gmail.com (E.V.H.); parisa.sinai@bristol.ac.uk (P.S.); Silvia.Cirillo@bristol.ac.uk (S.C.); 2Precision Immunology Institute, Icahn School of Medicine at Mount Sinai, New York, NY 10029, USA; 3Immunology Therapeutic Area, UCB, Slough, SL1 3WE, UK; Clare.Thompson@ucb.com (C.T.); farnaz.fallah-arani@ucb.com (F.F.-A.); 4R&D Oncology, AstraZeneca, Granta Park, Cambridge, CB21 6GH, UK; simon.dovedi@astrazeneca.com; 5Institute of Immunology and Immunotherapy, University of Birmingham, Birmingham, B15 2TT, UK

**Keywords:** regulatory T cell, tolerance, immunological synapse, central supramolecular activation cluster (cSMAC), supramolecular signaling complex, linker for activation of T cells (LAT), inhibitory receptors, CTLA-4, PD-1

## Abstract

Peripheral immune regulation is critical for the maintenance of self-tolerance. Here we have investigated signaling processes that distinguish T cells with regulatory capability from effector T cells. The murine Tg4 T cell receptor recognizes a peptide derived from the self-antigen myelin basic protein. T cells from Tg4 T cell receptor transgenic mice can be used to generate effector T cells and three types of T cells with regulatory capability, inducible regulatory T cells, T cells tolerized by repeated in vivo antigenic peptide exposure or T cells treated with the tolerogenic drug UCB9608 (a phosphatidylinositol 4 kinase IIIβ inhibitor). We comparatively studied signaling in all of these T cells by activating them with the same antigen presenting cells presenting the same myelin basic protein peptide. Supramolecular signaling structures, as efficiently detected by large-scale live cell imaging, are critical mediators of T cell activation. The formation of a supramolecular signaling complex anchored by the adaptor protein linker for activation of T cells (LAT) was consistently terminated more rapidly in Tg4 T cells with regulatory capability. Such termination could be partially reversed by blocking the inhibitory receptors CTLA-4 and PD-1. Our work suggests that attenuation of proximal signaling may favor regulatory over effector function in T cells.

## 1. Introduction

The random combinatorial generation of adaptive immune receptors by somatic recombination leads to emergence of substantial numbers of self-reactive B and T cells with a range of receptor affinities to self. While strongly self-reactive cells are eliminated by negative selection, the activation of adaptive immune cells with lower affinities for self needs to be suppressed by peripheral tolerance mechanisms to prevent autoimmune disease. Regulatory T cells (Treg) as defined by the transcription factor FoxP3 are a key element of peripheral tolerance [1]. They are generated during T cell development in the thymus through high affinity engagement of the TCR just below the threshold of negative selection as thymic Tregs [2,3] or upon activation of naïve T cells as enhanced by the cytokines TGFβ and IL-2 as peripheral Tregs [4,5]. Peripheral Treg generation can be recapitulated in vitro through activation of naïve T cells in the presence of TGFβ and high concentrations of IL-2 in the absence of other lineage-defining cytokines, leading to inducible Tregs (iTreg) [6]. All forms of Tregs are potent suppressors of autoimmune disease through a variety of mechanisms. The inhibitory receptor CTLA-4 is highly expressed by Tregs and can contribute to immune regulation [7,8], for example by stripping migratory dendritic cells of the costimulatory ligands CD80 and CD86 [9].

As an additional mechanism of peripheral immune regulation, repeated exposure of T cells to antigen can divert activated T cells to a regulatory phenotype, Tr1-like, characterized by secretion of IL-10 [10]. This can be applied therapeutically by injecting self peptides under non-immunogenic conditions in an escalating dose response [11,12]. Again, T cells therapeutically tolerized are characterized by high expression of inhibitory receptors [11]. In addition to strategies to promote in vitro or in vivo induction of regulatory and tolerized T cells to suppress unwanted immune responses, pharmacological treatments which foster induction of T cell tolerance are being explored. A potent inhibitor of the human mixed lymphocyte reaction targeting phosphatidylinositol 4-kinase IIIβ, UCB9608, can prevent the rejection of heterotopic allogeneic heart transplants in mice [13,14], establishing it as a pharmacological inducer of T cell tolerance. For a comparative understanding of signaling in different elements of T cell tolerance, we here investigate activation of iTregs, in vivo tolerized T cells and T cells treated with UCB9608.

In T cell activation, antigen specificity is mediated by recognition of peptide/MHC complexes on the surface of an antigen presenting cell (APC) by the T cell receptor (TCR). A common theme in T cell tolerance is that the same TCR can support immunity and tolerance, be it in the differentiation of naïve T cells into T effector versus regulatory T cells, in changing T cell function from immunity to tolerance upon repeated TCR engagement with antigen or in drug treatment to switch T cell function from immunity to tolerance. This raises two questions that we address here. What are signaling differences between immunogenic and tolerogenic T cells? Are there conserved signaling phenotypes in different types of T cells with regulatory capability? Signaling can be assessed at various levels. Commonly used are the posttranslational modification of signaling intermediates, such as their phosphorylation, or changes in gene expression. A particularly sensitive means to investigate signaling is the single cell determination of the formation and composition of supramolecular signaling structures in the activation of live T cells [15]. T cell signaling controls and is controlled by large subcellular structures: Prominent are F-actin based structures such as a peripheral F-actin ring to stabilize the interface with APCs [16,17] or an F-actin-based lamella to stabilize proximal signaling complexes [18]. Signaling intermediates can be segregated at the T cell distal pole [19] and brought together in supramolecular protein complexes, prominently at the center of the T cell/APC interface [20]. A supramolecular protein complex at the center of the T cell/APC interface, the central supramolecular activation cluster (cSMAC), is of particular interest as it is dynamically enriched in many proximal signaling intermediates [21]. Its formation is required during the first two minutes of T cell activation for efficient generation of IL-2 [15]. It is antagonized by a transient µm scale invagination to remove proximal signaling intermediates from the cellular interface [22]. Common for all of these structures is that they can be readily detected by large-scale live cell imaging as a function of time in single T cells activated by APCs and that their formation is highly sensitive to changes in T cell activation conditions [15,21]. A parallel determination of the dynamic formation of supramolecular signaling structures thus provides a sensitive fingerprint of T cell signaling.

Various types of tolerized or regulatory T cells display defects in signaling and its spatiotemporal organization. This is not surprising given the prominent expression of inhibitory receptors on such cells [7,11,23,24]. Tregs have a broadly dampened proximal signaling network [25]. T cells that continuously encounter their antigen in vivo display attenuated calcium signaling. Nevertheless, their transcriptional profile is more similar to that of effector T cells than that of Tregs [26]. Various aspects of T cell organization are impaired in tolerized or regulatory T cells [27,28,29,30,31,32]. What is missing is a study where signaling in various types of T cells with regulatory capability is directly compared. Here we use the Tg4 TCR transgenic system [33] to generate effector T cells, iTregs, in vivo tolerized T cells and T cells tolerized by treatment with the phosphatidylinositol 4-kinase IIIβ inhibitor UCB9608 [13]. Because all types of T cells in our experiments express the same Tg4 TCR, we can activate the T cells with the same APCs presenting the same peptide at the same concentration. Resulting changes in T cell activation are then not governed by different modalities of TCR engagement but are the consequence of different functional states. The formation of a supramolecular protein complex at the center of the T cell/APC interface was consistently less stable in the various forms of Tg4 T cells with regulatory capability. Stabilization of this complex with synthetic approaches or by blocking inhibitory receptors enhanced T cell function.

## 2. Materials and Methods

### 2.1. Mice

Tg4 [33] mice were bred in-house and maintained under SPF conditions with ad libitum access to water and standard chow at the University of Bristol. All animal experiments were carried out under the UK Home Office Project Licence number 30/2705 held by David C. Wraith and the study was approved by the University of Bristol Animal Welfare and Ethical Review Body. For immunotherapeutic tolerance induction, mice were treated s.c. every 3–4 days with 200 µL of MBP Ac1-9[4Y] in phosphate-buffered saline (PBS) in a dose escalation regime from 0.08 µg to 80 µg peptide in 4 treatments of 10-fold increases in peptide amount per treatment followed by two more treatments at the highest dose [11].

### 2.2. T Cell Culture

T cell cultures were set up from spleen or lymph node suspensions from Tg4 mice that had undergone in vivo tolerance induction (Ttol) or were left untreated (all other types of T cells). To generate Teff pep or Ttol cells, 4–5 × 10^6^ splenocytes or lymphocytes were incubated with 10 µg/mL MBP Ac1-9[4K]. Recombinant human IL-2 (TECIN, NCI Biological Resource Branch, Frederick, MD, USA) was added 24 h later at 50 U/mL. To generate iTreg pep cells, 4–5 × 10^6^ splenocytes or lymphocytes were incubated with 10 µg/mL MBP Ac1-9[4K] in the presence of 10 ng/mL TGFβ (Peprotech, London, UK) and 100 U/mL rhIL-2. Naïve T cells are commonly primed with antibodies instead of agonist peptide. To generate such T ab cells, CD4^+^ Tg4 T cells were purified from Tg4 splenocytes or lymphocytes with the MagniSort Mouse CD4 T cell Enrichment Kit (eBioscience, San Diego, CA, USA) according to the MagniSort Negative Selection Protocol III. Naïve CD4^+^ T cells were cultured in a 48-well plate (1.5 × 10^6^ per well) coated with 1 µg/mL anti-CD3ε clone 145-2C11 (eBioscience) and 2 µg/mL anti-CD28 clone 37.51 (eBioscience) for 48 h and then transferred to antibody-free wells. To generate Teff ab T cells, rhIL-2 was added at 24 h at 50 U/mL. iTreg ab T cells were cultured in the presence of 10 ng/mL TGFβ and 100 U/mL rhIL-2. T cells were used at 5–6 days of culture. To determine roles of inhibitory receptors in iTreg generation, the inhibitory receptors CTLA-4 and PD-1 were blocked for the entire duration of the iTreg culture with 10 µg/mL each of anti-CTLA-4 (clone 9D9, provided by AstraZeneca) and anti-PD-1 (clone RMP1-14, provided by AstraZeneca) antibodies.

### 2.3. Cell Lines

PL8 cells [34] are a B cell hybridoma expressing I-A^u^ MHC at high to intermediate levels. They were maintained in culture in RPMI with 10% FBS.

### 2.4. Flow Cytometry, Proliferation and Cytokine Measurement

For intracellular staining for Foxp3, the Foxp3/Transcription Factor Staining Buffer Set (eBioscience, San Diego, CA, USA) was used. Cells were fixed in Fixation/ Permeabilization reagent on ice for at least 30 min and then washed twice with Permeabilization buffer. Cells were resuspended in Permeabilization buffer and stained with anti-Fopx3 clone FJK-16s (1 µg/mL, eBioscience, San Diego, CA, USA) on ice for 30 min, washed and analyzed. For cell surface and intracellular staining of Teff pep and Ttol cells, cells were activated for 4 h at 37 °C with PMA and ionomycin at 5 ng/mL and 500 ng/mL, respectively, upon addition of Golgi Stop (BD Biosciences, San Jose, CA, USA) diluted to a final concentration of 1:1500. Cells were washed in ice cold PBS and stained with Fixable Viability Dye eFluor780 (eBioscience, San Diego, CA, USA) at 1 µL/mL in PBS for 20 min at room temperature. Cells were washed in PBS with 2 mM EDTA and 2% FBS and incubated with anti-mouse CD16/32 Fc block (eBioscience, San Diego, CA, USA) at 4 µg/mL for 5 min at room temperature. For extracellular staining, cells were incubated with fluorophore-conjugated antibodies, anti-LAG3 clone C9B7W (2 µg/mL) and anti-PD1 clone RPM1-30 (1 µg/mL, BioLegend, San Diego, CA, USA), for 30 min at 4 °C. For intracellular cytokine staining, cells were incubated with IC Fixation Buffer (eBioscience, San Diego, CA, USA) for 30 min at 4 °C, washed twice with Permeabilization Buffer (eBioscience, San Diego, CA, USA) and incubated with anti-IFNγ clone XMG1.2 (1 µg/mL, BD Biosciences, San Diego, CA, USA) or anti-IL-10 clone JES5-16E3 (1 µg/mL, BioLegend, San Diego, CA, USA) diluted into Permeabilization Buffer for 30 min at 4 °C, washed and analyzed.

For in vitro intracellular dye dilution proliferation assays, splenic CD4^+^ T cells were isolated from untreated Tg4 mice using a CD4^+^ T-cell isolation kit (Miltenyi Biotec, Bergisch Gladbach, Germany) and labelled with 5 µM CellTrace Violet (CTV) (Invitrogen Carlsbad, CA, USA). CTV-labelled responder CD4^+^ T cells (5 × 10^5^) were then cultured with 1 × 10^6^ irradiated B10.PL splenocytes as APC and 0.1 to 10 μg mL^−1^ MBP Ac1-9[4K]. CTV dilution was determined by flow cytometry on day 4.

To determine concentrations of IFNγ and IL-10 in tissue culture supernatants, supernatants were collected from cell culture by centrifugation and stored at −20 °C or used on the day. The OptEIA ELSIA kits from BD Biosciences (San Diego, CA, USA) were used according to the manufacturer’s protocol. Briefly, Nunc-Immuno MicroWell 96-Well Plates (Thermofisher, Waltham, MA, USA) were coated with 100 µL capture antibody diluted in coating buffer at 4 °C overnight. Wells were blocked with >200 µL assay diluent for 1 h at room temperature. 100 µL sample was incubated at room temperature for 2 h. Detection antibody and enzyme reagent were incubated for 1 h at room temperature. Duplicates or triplicates were applied.

### 2.5. Retroviral Transduction

Retroviral vectors for the expression of LAT-GFP, the DAG sensor (C1)_2_-GFP, TCRζ-GFP, the F-actin sensor F-tractin-GFP, LAT V3-GFP, and LAT Vav-GFP have been published [15,18]. Generation of MMLV-based recombinant retrovirus, retroviral infection of primary T cells at 48 h of culture and FACS sorting for sensor-GFP^+^ T cells at minimal sensor expression levels as close as possible to endogenous protein levels at day 5 of culture have been described in detail [21,35,36].

### 2.6. Live Cell Imaging

Live cell imaging of Tg4 T cells has been described in detail [36]. Briefly, PL8 cells were pre-loaded with 10 µg/mL MBP Ac1-9 [4Y] peptide for >4 h and combined with pre-sorted GFP^+^ T cells in a glass-bottom plate on the stage of a spinning disk microscope system (UltraVIEW 6FE system, Perkin Elmer, Waltham, MA, USA; DMI6000 microscope, Leica, Wetzlar, Germany; CSU22 spinning disk, Yokogawa, Tokyo, Japan). GFP data was collected as 21 z-sections at 1 µm intervals every 20 s at 37 °C in PBS containing 10% FCS, 1 mM CaCl_2_ and 0.5 mM MgCl_2_. To determine roles of inhibitory receptors, the inhibitory receptors CTLA-4 and PD-1 were blocked with 10 µg/mL each of anti-CTLA-4 (clone 9D9, provided by AstraZeneca) and anti-PD-1 (clone RMP1-14, provided by AstraZeneca) antibodies.

Images were analyzed with the Metamorph software (Molecular Devices, San Jose, CA, USA). Cell couples were identified using the differential interference contrast (DIC) bright field images as when either the T cell/APC interface had reached its full width or the cells had been in contact for 40 s, whichever came first. The subcellular localization of GFP-tagged protein sensors at each time point was classified. Interface enrichment of fluorescent proteins at less than 35% of the cellular background was classified as no accumulation. For enrichment above 35% six mutually exclusive interface patterns that reflect supramolecular subcellular structures were: accumulation in a large supramolecular protein complex at the center of the T cell:APC interface [15,21] (central), accumulation in a large presumably degradative T cell invagination [22] (invagination), accumulation that covered the cell cortex across central and peripheral regions (diffuse), accumulation in a broad actin-based interface lamella [18] (lamellal), accumulation at an F-actin ring at the periphery of the interface (peripheral) or in smaller membrane protrusions (asymmetric). Geometrical criteria for pattern classification are detailed in [21] and patterns are graphically illustrated in Appendix A.

For corroboration, some imaging data were also analyzed using computational methods [35,37,38] incorporated into CellOrganizer v2.8.1 (http://www.cellorganizer.org). Briefly, starting with the manual cell couple identification described above T cells were segmented, reoriented with the T cell:APC interface facing up using the most recent implementation of the one point method [38] and the cell shape was standardized to a half spheroid to allow voxel-by-voxel comparison across all cell couples analyzed. To measure interface enrichment, we defined an interface enrichment region as the 10% most fluorescent voxels of the average probability distribution across all cells and time points. We defined enrichment to be the ratio of the mean probability in the distribution of that sensor for that cell at that time point within the interface enrichment region and the mean probability in the entire cell. To measure enrichment at the interface center, we defined a small cylindrical “central core” region reaching from the interface center into the T cell: with the relative radius of the interface set to be “1” we used a relative radius and height (depth into the cell) of the central cylinder of “0.5”.

For the morphometric analysis we drew up to four lines in DIC images of T cell/APC couples at defined times relative to tight cell couple formation. The “interface diameter” connects the two outmost points at each edge of the interface. The “cell width” is a line at the widest part of the T cells perpendicular to a line connecting the center of the interface with its posterior end (see cell length). We used a combination of two criteria to determine whether a T cell has a scorable lamella, the existence of a “neck”, i.e., a point between the interface and the widest part of the cell where the cell diameter is smaller than that of the interface and the cell width, plus a distance of this neck from the interface of >1.3 µm. When such scorable lamella was formed we measured the “lamellal length” as the distance between interface and neck and the “cell length” as the distance between the interface and the posterior end of the T cell along the T cell midline. If the T cell was bent this line could have inflection points. The measurements are graphically illustrated in Appendix A. The “T cell shape factor” is calculated by dividing the lamellal length by the interface diameter.

## 3. Results

### 3.1. Tg4 T Cells as a Model to Investigate Signaling in T Cells with Regulatory Capability

Immune regulation by T cells is critical in tolerance and autoimmune disease. Therefore, we investigated signaling in T cells upon induction of regulatory function using the Tg4 mouse model of multiple sclerosis. Tg4 CD4^+^ T cells [33] recognize the acetylated N-terminal peptide of myelin basic protein Ac1-9[4K] and its high affinity MHC-binding analogue Ac1-9[4Y] as presented by I-A^u^. Tg4 T cells differentiated into Th1 or Th17 cells can mediate experimental autoimmune encephalitis in vivo [39]. To investigate signaling upon Tg4 T cell effector function as reference data for the regulatory work, we generated Tg4 effector T cells in vitro by priming naïve Tg4 lymphocytes with Ac1-9[4K] and subsequently expanding them in 50 U/mL IL-2 (“Teff pep”). The formation of supramolecular signaling structures as determined by live cell imaging is a sensitive means to investigate signaling. To visualize such structures, we retrovirally transduced in vitro primed Tg4 T cells for the expression of GFP-tagged signaling intermediates and FACS-sorted GFP^+^ Tg4 Teff pep cells after five days of tissue culture as established [35] (Figure 1A).

To investigate T cells with regulatory capability, we used three experimental strategies. We generated inducible regulatory T cells by priming naïve Tg4 lymphocytes with Ac1-9[4K] in the presence of 10 ng/mL TGFβ and 100 U/mL IL-2 (‘iTreg pep’) (Figure 1A) [40]. iTreg pep cell cultures were mostly >70% FoxP3^+^ (Figure 1B and Appendix A) and not used unless >75% FoxP3^+^. Retroviral transduction to express GFP-tagged signaling sensors did not affect induction of FoxP3 (Appendix A). Consistent with regulatory function the balance of cytokine secretion was shifted from IFNγ to IL-10 as compared to Teff pep cells during iTreg pep induction (Figure 1C) and re-stimulation (Figure 1D). While Teff pep cells consistently secreted more IFNγ than iTreg pep cells during a time course of in vitro differentiation (*p* < 0.001), iTreg pep cells started to secret more IL-10 without reaching statistical significance though at day 5, one day after a plateau of maximal FoxP3 expression was reached as described later.

We generated FoxP3^–^ Tg4 T cells with regulatory capability in vivo using an established dose escalation protocol of repeated injection of mice with Ac1-9[4Y] under non-immunogenic conditions [11]. We in vitro expanded and retrovirally transduced such cells as done in the generation of Teff pep cells to generate tolerized Tg4 T cells (‘Ttol’) (Figure 1A). Consistent with preservation of a regulatory phenotype upon in vitro tissue culture, IL-10 expression was elevated in Ttol but not expression of IFNγ, again independent of retroviral transduction (Figure 1E and Appendix A). Expression of the inhibitory receptors Lag3 and Pd-1 was elevated (Figure 1E).

Finally, in contrast to the induction of regulatory capacity over days in the iTreg pep and Ttol cells, we acutely treated Tg4 Teff pep cells with 100 nM of the phosphatidylinositol 4-kinase IIIβ inhibitor UCB9608 (‘Teff iPI4K’). Treatment of mice with UCB9608 mediates tolerance in heart transplantation [13]. UCB9608 inhibited the proliferation of Teff cells, i.e., induced anergy (Figure 1F and Appendix A), and enhanced FoxP3 induction in the generation of iTregs (Figure 1G) at concentrations above 50 ng/mL (Appendix A). In vivo UCB9608 reduced T cell numbers, yet the fraction of T cells expressing FoxP3, the inhibitory receptors Tigit and Tim3 and IL-10 was increased in the remaining cells as to be described in a future manuscript. Treatment with UCB9608 thus promoted a regulatory phenotype.

This Tg4 experimental system allows an effective comparative investigation of signaling in T cells with regulatory capability. Teff, iTreg, Ttol and Teff iPI4K cells all express the same Tg4 TCR. By activating them with the same PL8 B cell lymphoma APCs incubated with the same amount (10 µg/mL) of the same Ac1-9[4Y] agonist peptide all T cells receive an identical TCR stimulus. Signaling differences could therefore be attributed to their divergent differentiation state.

### 3.2. LAT Accumulation at the Center of the T Cell APC Interface Was More Rapidly Terminated in T Cells with a Regulatory Phenotype

Accumulation of LAT at the center of the T cell APC interface in a cluster of supramolecular signaling complexes within the first two minutes of T cell activation promotes efficient T cell activation [15,21]. LAT accumulation in a large invagination at the interface center during the same time, however, is thought to initiate the termination of signaling activity [22]. Distinguishing these opposing signaling events, the cluster of supramolecular signaling complexes remains at the T cell APC interface while the invagination reaches µm deep into the T cells. We imaged LAT accumulation in Tg4 T cells (Figure 2A, Appendix A). LAT accumulated effectively at the interface center in the activation of Teff pep cells (Figure 2B, Appendix A) reaching 60 ± 7% of cell couples with interface accumulation in any pattern at 20s after the formation of a tight cell couple with 42 ± 7% at the interface center and 13 ± 5% in the invagination. Central LAT accumulation varied in Tg4 T cells with a regulatory phenotype. It was almost completely lost in iTreg pep cells, not exceeding 9% of cell couples with central accumulation at any time, partially lost in Ttol cells, briefly peaking at 25 ± 6% of cell couples with central accumulation at the time of tight cell coupling, and less well sustained in Teff iPI4K cells, dropping to less than 10% of cell couples with central accumulation at 60 s after tight cell coupling (Figure 2B,C and Appendix A). However, the balance of LAT accumulation between the central and invagination patterns was consistently shifted towards the invagination (Figure 2D). In Teff pep cells central accumulation was more prominent than accumulation in the invagination across all time points within the first two minutes of tight cell coupling, the peak of LAT interface accumulation. In contrast, invagination accumulation was more prominent from 20 s to 2 min for all three Tg4 T cell types with regulatory capability. A rapid transition into the invagination, a likely degradative signaling structure, is indicative of ill sustained LAT accumulation in a cluster of supramolecular signaling complexes at the interface center.

iTregs are commonly generated by priming purified CD4^+^ T cells with agonist anti-CD3 plus anti-CD28 in the presence of TGFβ and high concentrations of IL-2. As this method is favored in clinical therapy settings, it is useful to also understand the function and phenotype of antibody-stimulated iTreg. LAT interface accumulation during the first two minutes of cell coupling, the time of maximal central LAT accumulation, in such antibody-induced iTreg (“iTreg ab”) was comparable to that in iTreg differentiated by stimulation with APC and agonist peptide (Appendix A). Sustained LAT accumulation in any pattern was more prominent in antibody as compared to peptide-primed Teff and iTreg cells, as not further pursued here. The pattern-based image analysis predominantly employed here is particularly powerful in distinguishing spatially similar signaling distributions such as central accumulation versus invagination versus accumulation in vesicles as commonly seen for LAT. A recently developed computational image analysis routine [37,38] provides superior quantitation of the intensity of signaling accumulations at the expense of diminished spatial resolution. Using this computational image analysis routine, we found diminished LAT accumulation at the interface and its center in iTreg pep as compared to Teff pep cells (Appendix A), thus corroborating the pattern-based analysis. Together the LAT data establish that acquisition of a regulatory phenotype consistently leads to the accelerated termination of LAT accumulation at the interface center, suggesting more transient and limited signaling activity.

### 3.3. cSMAC Formation Was Diminished upon Induction of a Regulatory Phenotype

LAT accumulation at the interface center is part of the formation of the central supramolecular activation cluster (cSMAC) [15]. In particular within the first two minutes of T cell activation a large number of proximal T cell signaling events associate with the cSMAC as required for efficient T cell activation [15]. To provide some insight into wider defects in cSMAC formation, we investigated interface accumulation of two signaling intermediates in some of the three models of Tg4 T cells with regulatory capacity. A key proximal signaling event in T cell activation is the generation of diacylglycerol (DAG). In Ttol cells interface accumulation of DAG as measured with a tandem C1 domain-GFP sensor was substantially diminished (Figure 3A, Appendix A). In comparison to Teff pep cells, DAG interface accumulation in Tg4 Ttol cells was reduced more rapidly after similar initial accumulation at the time of tight cell couple formation, leading to significantly (*p* < 0.05) reduced interface accumulation from 40 s to 180 s after tight cell coupling. As DAG is rapidly diffusible diminished DAG accumulation was extended across the entire interface.

Effective early cSMAC formation promotes subsequent clustering of the TCR at the interface center [21]. Consistent with inefficient cSMAC formation, central TCR accumulation five to seven minutes after tight cell coupling was substantially reduced in Ttol and iTreg pep (Figure 3B, Appendix A) as confirmed in iTreg ab cells (Appendix A). While central TCRζ-GFP accumulation reached 25 ± 7% of cell couples during that time in Teff pep cells it stayed below 5% and 10% in Ttol and iTreg pep cells, respectively. Together (Figure 4) these data suggest that overall cSMAC formation and associated signaling activity were diminished in Tg4 T cells with regulatory capacity.

As further discussed below, the Tg4 TCR is self-reactive. Central accumulation of the TCR has been shown to be diminished in the activation of self-reactive CD4^+^ T cells [31]. It is, therefore, of interest that central accumulation of all proteins investigated here in Tg4 Teff cells is consistently less extensive than that seen in the activation of foreign-reactive 5C.C7 and DO11.10 T cells by B cell lymphoma APCs and a high concentration of agonist peptide, comparable to the activation conditions used here [15,41].

### 3.4. Actin-Driven Formation of a Polarized Cell Couple Was Impaired upon Induction of a Regulatory Phenotype

Efficient early cSMAC formation requires rapid F-actin turnover [37]. One indicator of efficient F-actin turnover is the rapid recruitment of F-actin to the T cell/APC interface upon cell coupling. Such recruitment was delayed in iTreg pep, Ttol and Teff iPI4K cells (Figure 5A–C, Appendix A). While in Teff pep cells interface F-tractin-GFP [16] accumulation in any pattern reached 86 ± 4% already at the time of tight cell coupling, such accumulation did not exceed 85% until 60 s, 180 s, or ever in Ttol, Teff iPI4K or iTreg pep cells, respectively, as corroborated in iTreg ab cells (Appendix A). Early F-actin dynamics are required to first extend and then rapidly shorten initial often-long cell-wide lamellal sheets that the T cell uses to make contact with the APC. In comparison to Teff pep cells, iTreg pep cells displayed an increased frequency of distinct lamellae within the first minute of cell coupling with increased length relative to the interface diameter (Appendix A), indicative of impaired lamellal shortening. In contrast, Ttol and Teff iPI4K cells displayed shorter and less frequent lamellae (Appendix A), consistent with impaired lamellal extension. Both phenotypes, impaired extension and shortening of early lamellae are consistent with slowed F-actin turnover. Slowed early F-actin dynamics are in turn consistent with the less efficient formation and/or more rapid termination of cSMAC signaling.

At later time points the formation of smaller off-interface lamellae is an indication of impaired cell couple maintenance [38,42]. The frequency of cell couples with such off-interface lamellae within the first 7 min after tight cell coupling was increased from 58 ± 4% in Teff pep cells to 70 ± 5%, 73 ± 2% and 88 ± 2% in Teff iPI4K, Ttol, and iTreg pep cells (Figure 5D), respectively. Again, the iTreg pep cells displayed the most severe defect. Such later F-actin defects are indicative of consistently impaired F-actin dynamics upon induction of a regulatory phenotype.

F-actin dynamics not only drive cell couple polarization, they can also directly support signaling through the formation of an F-actin sheet with embedded smaller signaling complexes, in particular within the first two minutes of cell coupling [18]. We observed a shift in the pattern of F-actin interface accumulation from some lamellal/diffuse accumulation in Teff pep cells to almost exclusively peripheral patterning in both Ttol and Teff iPI4K cells (Figure 5B). The combined diffuse and lamellal accumulation of F-tractin in Teff pep cells stayed at or above 40% of cell couples during the entire first minute of tight cell coupling, indicative of substantial formation of a lamellal F-actin network. It did not exceed 10% in Ttol and Teff iPI4K cells and stayed around 20% in iTreg pep cells during the same period of time (Figure 5C). While details varied, F-actin dynamics in rapid F-actin interface recruitment, lamellal extension and retraction, off-interface lamellae and the formation of a signaling-sustaining F-actin sheet were consistently impaired in T cells with regulatory capability.

In summary of the imaging experiments to investigate the formation of supramolecular structures in Tg4 T cell activation, the induction of a regulatory phenotype consistently led to less efficient T cell polarization, in early and sustained F-actin dynamics as associated with diminished formation of supramolecular signaling structures, most prominently the cSMAC but also the F-actin-based lamellae (Figure 4; Figure 5E). The extensive scope of these defects is indicative of broadly attenuated signaling activity upon induction of regulatory capability.

### 3.5. cSMAC Formation Could Be Partially Restored with Synthetic Approaches in Tg4 iTreg Pep Cells

The cSMAC consists of multiple supramolecular signaling complexes [15]. Their formation is diminished upon attenuation of T cell stimulation [15]. Supramolecular signaling complexes require a network of multi-valent interactions for their formation [43,44]. Therefore, diminished formation is likely caused by a reduced valence in the associated signaling intermediates, e.g., through reduced tyrosine phosphorylation that is required for many phospho-tyrosine SH2 domain interactions in proximal T cell signaling. This reduction in valence in cSMAC formation can be overcome by adding protein interaction domains to LAT [15]. To investigate whether the impaired cSMAC formation upon induction of a regulatory phenotype is consistent with this model of cSMAC function, we expressed two LAT variants with increased valence in Tg4 iTreg pep cells, LAT fused to the V3 domain of PKCθ (‘LAT V3′) and LAT fused to the Vav1 SH3-SH2-SH3 domain module (‘LAT Vav’) [15]. The V3 domain of PKCθ and the Vav1 SH3-SH2-SH3 domain module drive sustained central accumulation and central accumulation during the first minute of cell coupling, respectively. Overall iTreg pep cell LAT interface accumulation was greatly increased upon expression of LAT V3 and LAT Vav reaching or even exceeding levels seen in Teff pep cells (Figure 6A,B, Appendix A) as corroborated in iTreg ab cells and by computational image analysis (Appendix A). While LAT accumulation in any interface pattern did not exceed 30% of cell couples in iTreg cells expressing LAT-GFP (Figure 2B), it reached 80 ± 5% and 63 ± 7% upon expression of LAT V3-GFP and LAT Vav-GFP, respectively (Figure 6B). However, the LAT accumulation at the interface center was only partially restored (Figure 6C), even though the balance of central LAT accumulation to that in the invagination reached the same level as seen in Teff pep cells (Figure 6D). While accumulation in the invagination was consistently more prominent than that in the cSMAC in iTreg pep T cells expressing LAT-GFP throughout the first minute of cell coupling, the opposite occurred for iTreg pep cells expressing LAT V3-GFP or LAT Vav-GFP, comparable to Teff pep cells (Figure 6D). Consistent with a close mutual connection between early cSMAC formation and F-actin dynamics, the ability of iTreg pep cells to shorten the initial lamellal sheet used to make contact with APCs was partially restored upon expression of LAT V3 and LAT Vav (Appendix A).

In the work on costimulation blockade in 5C.C7 Teff cells, restoration of diminished IL-2 generation upon costimulation blockade as a key T cell effector function required rebuilding the cSMAC to levels very similar to those in T cell receiving a full stimulus, not more, not less [15]. To investigate whether such restoration equally applies to Tg4 T cells, we determined cytokine secretion in iTreg cells expressing LAT-V3 or LAT-Vav. Upon expression of LAT V3-GFP secretion of IL-10 and IFNγ was diminished (Figure 6E), consistent with reduced IL-2 generation in 5C.C7 CD4^+^ Teff cells upon similarly excessive reconstitution of LAT clustering with LAT V3 upon costimulation blockade [15]. Upon expression of LAT Vav-GFP, however, we observed a trend towards more efficient IL-10 secretion, with no effect on IFNγ (Figure 6E), a trend comparable to restoration of IL-2 generation in 5C.C7 T cells expressing LAT-Vav upon costimulation blockade. Consistent with such work, restoration of LAT clustering in Tg4 iTreg cells with LAT Vav more closely resembled Teff pep cells than restoration with LAT V3. However, in contrast to the 5C.C7 T cell work, such restoration still left substantial differences, i.e., excessive LAT accumulation in any pattern and incomplete restoration of central accumulation. This only partial restoration may explain the more limited functional effect in Tg4 as compared to 5C.C7 T cells. Nevertheless, together (Figure 4) these data support the notion that valence-driven early cSMAC formation is an amplifier of T cell effector function also in Tg4 T cells, including a reduction upon induction of a regulatory phenotype.

### 3.6. Pd-1 and Ctla-4 Impaired cSMAC Formation upon Induction of a Regulatory Phenotype

The identification of molecular regulators of diminished proximal signaling upon induction of regulatory capability is of interest. Regulatory T cells, including Tg4 iTreg, Ttol and Tg4 T cells from mice treated with UCB9608, displayed increased expression of inhibitory receptors (Figure 1E) [7,11,23,24]. We, therefore, determined the role of two of the most prominent inhibitory receptors, Pd-1 and Ctla-4, on the signaling of iTreg pep cells using blocking antibodies. Blockade of iTreg pep cells with α-Pd-1 plus α-Ctla-4 partially restored early central LAT accumulation and partially overcame the rapid movement of LAT into the invagination (Figure 7A–C, Appendix A). While accumulation in the invagination was consistently more prominent than that in the cSMAC in iTreg pep T cells throughout the first minute of cell coupling, the opposite occurred upon blockade of Pd-1 and Ctla-4, more comparable to Teff pep cells (Figure 7C). Partial restoration of central LAT accumulation was confirmed in iTreg ab cells with a slightly more pronounced effect of α-Ctla-4 as opposed to α-Pd-1 and without substantial synergy between the treatments (Appendix A). Computational analysis confirmed partial reconstitution of overall LAT interface accumulation (Appendix A).

Consistent with a role of early cSMAC formation in generating signals necessary for later TCR accumulation, TCR accumulation at the interface center was restored in iTreg pep cells upon blockade of Pd-1 and Ctla-4 (Figure 7D, Appendix A). Central accumulation of TCRζ-GFP reached 49 ± 8% of cell couples 7 min after cell coupling in iTreg pep cells upon blockade of Pd-1 and Ctla-4, significantly (*p* < 0.001) exceeding such accumulation in control treated iTreg pep cells (5 ± 3%) and more comparable to Teff pep cells (25 ± 7%). Again, linking the cSMAC to F-actin dynamics, the ability of iTreg pep cells to shorten the initial lamellal sheet used to make contact with APCs was partially restored upon blockade of Pd-1 and Ctla-4 (Appendix A) with little effect on overall F-actin accumulation (Appendix A). These data establish the inhibitory receptors Ctla-4 and Pd-1 as consistent regulators of the spatiotemporal organization of T cell signaling, in cSMAC formation, subsequent TCR clustering and F-actin dynamics. The effect of the antibodies is likely caused by the direct blockade of ligand binding of Ctla-4 and Pd-1 by the antibodies rather than by steric interference with synapse organization by the antibody/inhibitory receptor complexes, as dextran with a diameter of 13 nm, larger than an antibody, can readily move through the interface between a Jurkat T cell with a Raji APC [45].

As a process on a different timescale than the re-activation of iTreg cells investigated in our imaging experiments, blocking Pd-1 and Ctla-4 also interfered with the induction of iTreg cells. Blockade of Pd-1 and Ctla-4 led to a substantial reduction of the percentage of FoxP3^+^ cells upon differentiation of Tg4 splenocytes in the presence of TGFβ and IL-2 (Appendix A) from 66 ± 5% of Tg4 T cells with expression of FoxP3 at day 6 to 45 ± 5% with synergy between α-Pd-1 and α-Ctla-4 (Appendix A). This was accompanied by significantly (*p* < 0.01) enhanced IL-10 secretion during the in vitro T cell differentiation in the presence of TGFβ and IL-2 (Appendix A). While not further pursued here, the attenuated signaling in T cells with regulatory capability may thus extend to their induction.

## 4. Discussion

Here we have investigated supramolecular signaling structures in the activation of T cells with a regulatory capability. By using the Tg4 TCR transgenic system [33] to generate effector T cells, iTregs, in vivo tolerized T cells and T cells acutely tolerized by treatment with UCB9608 we can activate the T cells with the same PL8 B cell lymphoma APCs presenting the same MBP Ac1-9[4Y] agonist peptide at the same concentration of 10 µg/mL. Resulting changes in T cell signaling are then not governed by different modalities of TCR engagement but are the consequence of different functional states of the T cells. We thus found that proximal signaling as determined through the formation of supramolecular signaling structures was consistently attenuated in T cells with regulatory capability. In comparing Tg4 data to those generated using other primary TCR transgenic T cells, one needs to consider that the Tg4 TCR is self-reactive. Self-reactive T cells show diminished clustering of their TCR at the interface center [31]. Comparing the Tg4 Teff pep data to data from the activation of T cells expressing the foreign-reactive TCRs 5C.C7 or DO11.10 with B cell lymphoma APCs and a high concentration of agonist peptide [15,18,21,46], comparable to the T cell activation conditions used here, central accumulation of LAT, DAG and TCRζ was diminished moderately in the Tg4 Teff pep cells. These data thus confirm the prior work and extend it to other signaling intermediates. Nevertheless, faster termination of central signaling accumulation was a principal signaling defect in Tg4 T cells with regulatory capability established here. In 5C.C7 T cells central signaling accumulation is indicative of the formation of a cluster of membrane-anchored supramolecular protein complexes [15]. Central signaling accumulation in Tg4 T cells also most likely represents the formation of supramolecular protein complexes: Such formation critically depends on the valence of complex components [44,47]. Accordingly, expression of LAT variants with addition of protein interactions domains to increase LAT valence enhanced central signaling localization in Tg4 iTreg pep, similar to 5C.C7 T cells upon costimulation blockade [15]. Our findings raise a number of questions.

What are the molecular regulators of diminished central signaling? Tg4 iTreg pep and Ttol cells express increased amounts of inhibitory receptors. Blocking two of them, CTLA-4 and PD-1, at least partially restored central signaling localization in Tg4 iTreg pep cells (Figure 7) implying inhibitory receptors as regulators of central signaling. CTLA-4 and PD-1 are well placed to do so through their connections to costimulatory signaling. CD28 engagement accelerates F-actin turnover and thus contributes to central LAT accumulation [37]. CTLA-4 directly competes with CD28 for CD80/CD86 ligand engagement [9,48,49,50]. PD-1 inhibits the activity of the critical costimulatory signaling intermediate Akt [51]. The delay in the formation of a peripheral F-actin ring in T cells with a regulatory capability (Figure 5) is consistent with a costimulation-driven actin-dependent regulation of central signaling localization in Tg4 T cells. As an alternate costimulation-linked signaling event, accumulation of the E3 ubiquitin ligase Cbl-b at interface between T cells and an activating cover slip as an APC substitute is enhanced in in vivo tolerized Tg4 T cells [32], as also seen in the activation of anergic CD4^+^ T cells [29]. Beyond costimulation, the suggested recruitment of the tyrosine phosphatase SHP-2 to PD-1 [52] could directly interfere with tyrosine phosphorylation in TCR signal transduction, consistent with diminished recruitment of ZAP-70 to the interface between in vivo tolerized Tg4 T cells and APCs [32]. It remains unclear whether acute induction of regulatory capability through treatment with UCB9608 is related to inhibitory receptors. Given that CTLA-4 is largely stored in vesicles [53], inhibition of phosphatidylinositol 4-kinase IIIβ could affect insertion of CTLA-4 into or removal from the plasma membrane during T cell activation. While suppressed costimulatory signaling in T cells with regulatory capability is consistent with much of our data, future work will have to conclusively resolve the mechanism of diminished central signaling.

What is the time scale of the induction of signaling changes that are associated with a regulatory state? Cell signaling can be regulated on the time scale of seconds to minutes, for example through post-translational modifications. Cell signaling can be regulated on the time scale of days, for example through changes in the expression of signaling intermediates. The times scales of the induction of signaling changes underpinning a regulatory state are of interest as they are likely linked to how quickly regulatory states can be reversed again. Differentiation of T cells into Tregs and in vivo tolerization of Tg4 T cells leads to transcriptional changes as controlled by epigenetic modifications [32,54,55]. At least some of the signaling changes observed here thus are likely induced and maintained on the time scale of days. Nevertheless, acute treatment of Tg4 Teff pep cells with UCB9608 induced regulatory signaling features on the time scale of minutes and bypassing proximal signaling in in vivo tolerized Tg4 T cells with phorbol ester and a calcium ionophore can reverse consequences of transcriptional changes [32]. We, therefore, expect that long and short-term adaptations synergize in the induction of regulatory states. As central signaling was more extensively diminished in Tg4 iTreg and Ttol cells than in Teff iPI4K cells, we also expect that long-term adaptation leads to a more profound and, therefore, possibly more stable establishment of a regulatory state.

What is the functional significance of diminished central signaling in T cells with regulatory capability? Diminished central signaling could be critical in diverting T cell activation from effector to regulatory responses. Alternatively, it could simply attenuate all aspects of T cell activation. Enhanced secretion of IFNγ and IL-10 upon restoration of central signaling organization in Tg4 iTreg pep cells by expression of LAT variants with additional protein interaction motifs (Figure 6) or during T cell differentiation by blocking inhibitory receptors (Appendix A) suggest a general attenuation of T cell effector function in T cells with regulatory capability. What purpose could that serve? Regulatory T cells are not limited to suppressive functionality. They can revert to the execution of effector functions of conventional T cells, such as the secretion of pro-inflammatory cytokines [56]. Signaling may contribute to this balance. Some effector functions in regulatory T cells don’t require strong T cell signal transduction, such as high expression of CD25 to allow effective competition for IL-2 [57,58] or high CTLA-4 expression to compete with costimulation [9,59]. We speculate that a general attenuation of T cell signaling in T cells with regulatory capability serves to shift the balance from effector functions that are most prominent upon strong T cell signaling to regulatory function that is independent of strong signaling.

## Figures and Tables

**Figure 1 cells-10-00418-f001:**
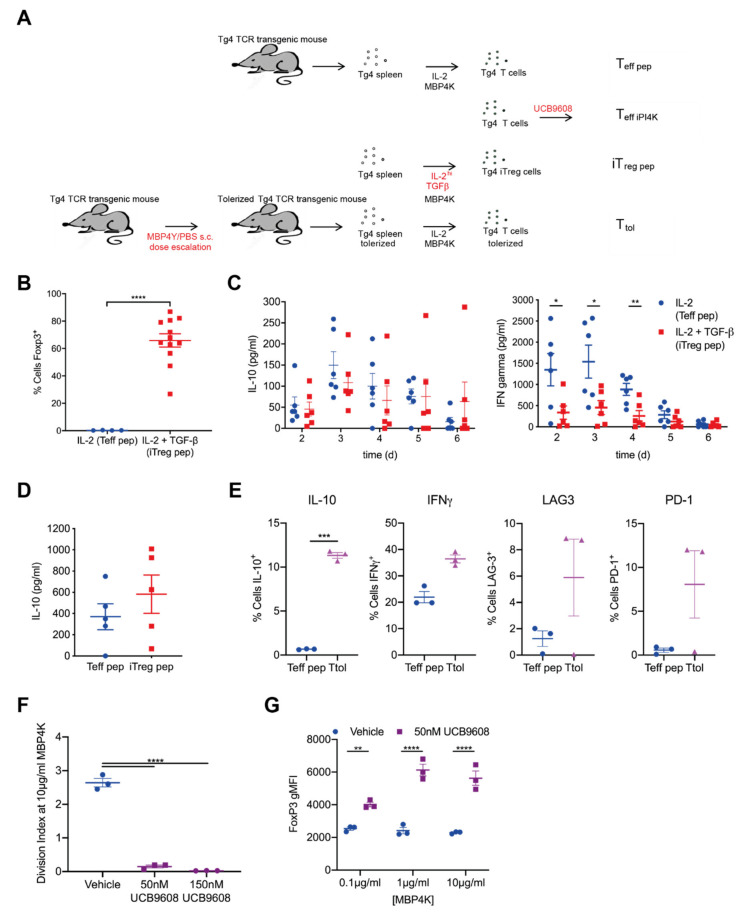
Three experimental models of T cells with regulatory capability. (**A**) Schematic of Tg4 T cell culture to generate Teff pep, iTreg pep, Ttol and Teff iPI4K cells; (**B**) Tg4 splenocytes were incubated with 10 µg/mL MBP Ac1-9[4K] peptide for 5 days in the presence of 50 U/mL IL-2 or 10 ng/mL TGFβ and 100 U/mL IL-2 and stained for CD4 and FoxP3. Data are expressed as % FoxP3^+^cells (n = 4–12 independent experiments). **** *p* < 0.0001; (**C**) Tg4 splenocytes were incubated with 10 µg/mL MBP Ac1-9[4K] peptide for 6 days in the presence of 50 U/mL IL-2 or 10 ng/mL TGFβ plus 100 U/mL IL-2 and IFNγ and IL-10 amounts in tissue culture supernatants were determined by ELISA. Data are expressed as amount of cytokine (n = 5 independent experiments). * *p* < 0.05, ** *p* < 0.01; (**D**) Tg4 eff pep or iTreg pep cells were re-activated with PL8 APCs (10 µg/mL Ac1-9[4Y]) for 18 h and IL-10 amounts were determined by ELISA. Data are expressed as amount of cytokine (n = 6 independent experiments); (**E**) Tg4 T cells were tolerized in vivo with a dose escalation of MBP Ac1-9[4K] peptide in PBS or left untreated, incubated in vitro with 10 µg/mL MBP Ac1-9[4K] peptide for 5 days in the presence of 50 U/mL IL-2, retrovirally transduced to express GFP, re-activated with PMA and Ionomycin and stained for LAG-3, PD-1, intracellular IL-10 and IFNγ. Data are expressed as % T cells positive for the indicated marker (n = 3). * *p* < 0.05, ** *p* < 0.01, *** *p* < 0.001; (**F**) Naïve Tg4 T cells were labelled with the cell proliferation dye CellTrace Violet and incubated with irradiated splenocytes and the indicated concentrations of MBP Ac1-9[4K] peptide for 4 days in the presence of 50 U/mL IL-2 with the indicated concentration of UCB9608 or vehicle. Data are expressed as the average number of cell divisions (‘division index’) at an MBP Ac1-9[4K] peptide concentration of 10 µg/mL. **** *p* < 0.0001; (**G**) Tg4 splenocytes were incubated with the indicated concentrations of MBP Ac1-9[4K] peptide for 6 days in the presence of 50 U/mL IL-2 with 50 nM UCB9608 or vehicle and stained for FoxP3. Data are expressed as MFI of FoxP3^+^ (n = 3). ** *p* < 0.01, **** *p* < 0.0001.

**Figure 2 cells-10-00418-f002:**
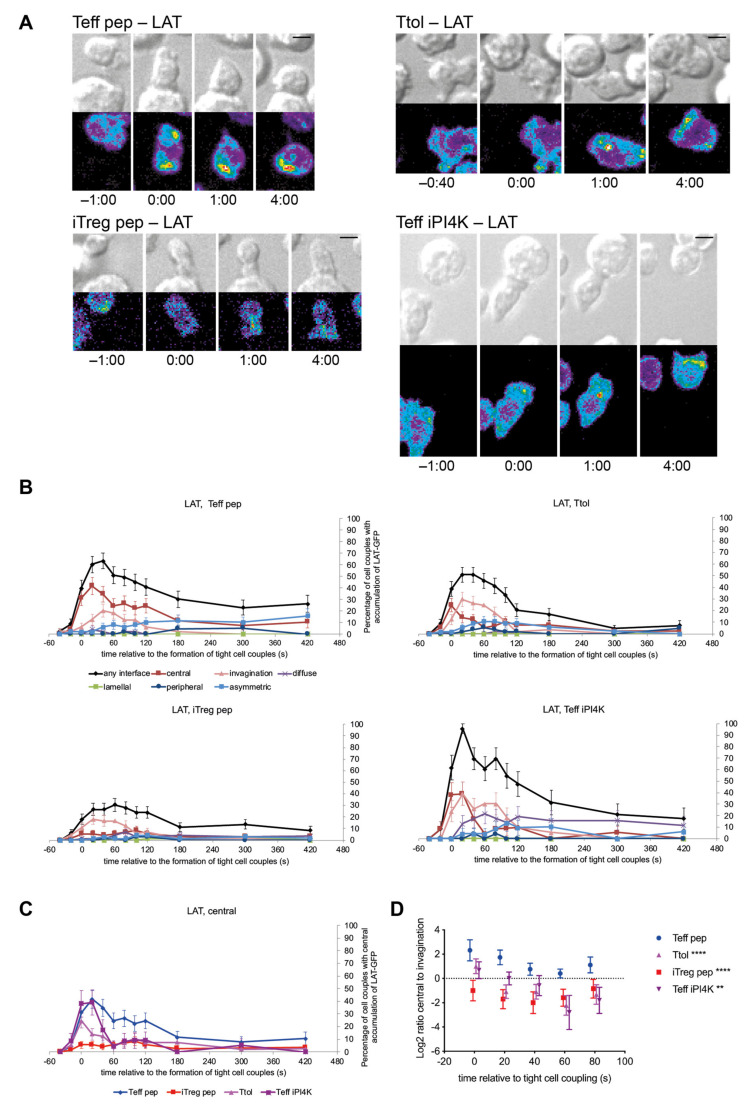
LAT association with the cSMAC is terminated more rapidly in T cells with regulatory capability. Tg4 T cells as indicated transduced to express LAT-GFP were activated with PL8 APCs (10 µg/mL Ac1-9[4Y]). (**A**) Representative LAT-GFP imaging data. Differential interference contrast (DIC) images are shown in the top row, with top-down, maximum projections of 3-dimensional LAT-GFP fluorescence data in the bottom row. LAT-GFP fluorescence intensities are displayed in a rainbow-like false-color scale (increasing from blue to red). The scale bar corresponds to 5 µm. Corresponding videos are available as Videos S1–S4. (**B**,**C**) Data are expressed as the percentage of cell couples with LAT-GFP accumulation in the indicated patterns (Appendix A) relative to tight cell couple formation. Number of cell couples analyzed are n = 49, 57, 72, 23 for Teff pep, Ttol, iTreg pep, Teff 9608 cells, respectively, from 2–5 independent experiments. Statistical significance of differences between conditions is given in Appendix A; (**B**) all patterns for each cell type separately; C only central accumulation in direct comparison between the cell types; (**D**) The same data are expressed as the log_2_ of the ratio of percent Tg4 T cells with accumulation in the central over the invagination pattern relative to tight cell couple formation. Values above 0 indicate preferential central accumulation, below 0 preferential accumulation in the invagination. Data points are slightly nudged to increase legibility. ** *p* < 0.01, **** *p* < 0.0001 vs. Teff pep.

**Figure 3 cells-10-00418-f003:**
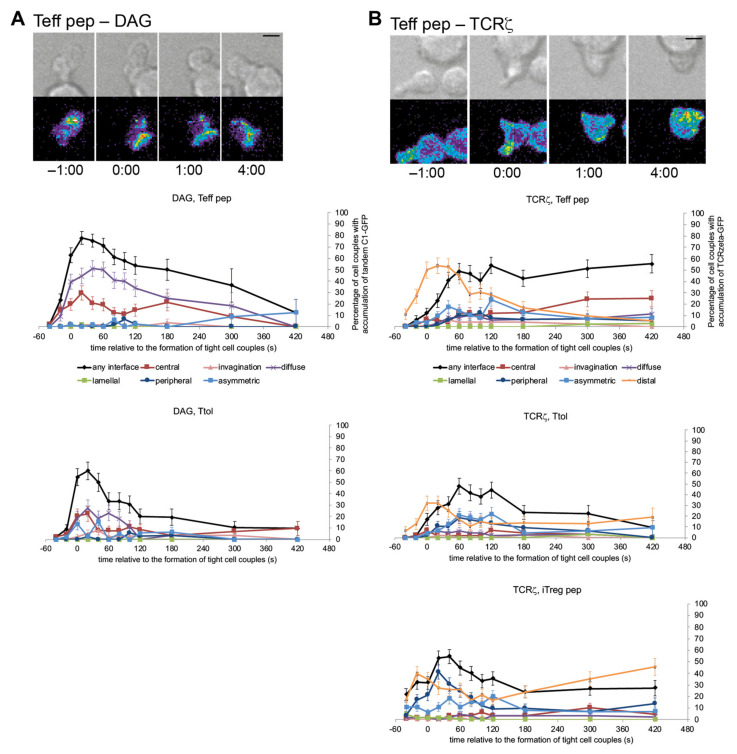
cSMAC formation was diminished but not abolished upon induction of a regulatory phenotype. (**A**) Tg4 T cells as indicated transduced to express Tandem C1-GFP as indicated were activated with PL8 APCs (10 µg/mL Ac1-9[4Y]). On the left, representative tandem C1-GFP imaging data as in Figure 2A. A corresponding video is available as Video S5. On the right, data are expressed as the percentage of cell couples with DAG accumulation measured with tandem C1-GFP in the indicated patterns (Appendix A) relative to tight cell couple formation. Number of cell couples analyzed are n = 56, 45 for Teff pep and Ttol cells, respectively, from 1 independent experiment each. Statistical significance of differences between conditions is given in Appendix A. (**B**) Tg4 T cells as indicated transduced to express TCRζ-GFP were activated with PL8 APCs (10 µg/mL Ac1-9[4Y]). On the top left, representative TCRζ-GFP imaging data as in Figure 2A. A corresponding video is available as Video S6. Data are expressed as the percentage of cell couples with TCRζ-GFP accumulation in the indicated patterns (Appendix A) relative to tight cell couple formation. Number of cell couples analyzed are n = 52, 48, 66 for Teff pep, Ttol, iTreg pep cells, respectively, from 3–5 independent experiments. Statistical significance of differences between conditions is given in Appendix A.

**Figure 4 cells-10-00418-f004:**
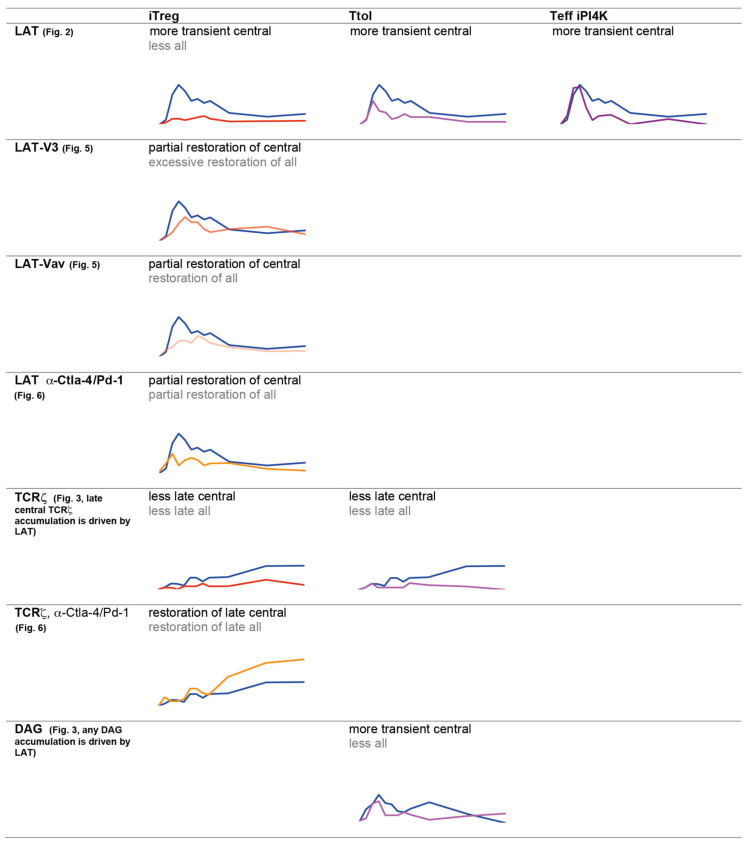
Summary of central accumulation phenotypes. Changes in accumulation phenotypes of the different Tg4 cell types with regulatory capability in comparison to Teff cells are summarized across all experiments with a focus on central accumulation of the manuscript.

**Figure 5 cells-10-00418-f005:**
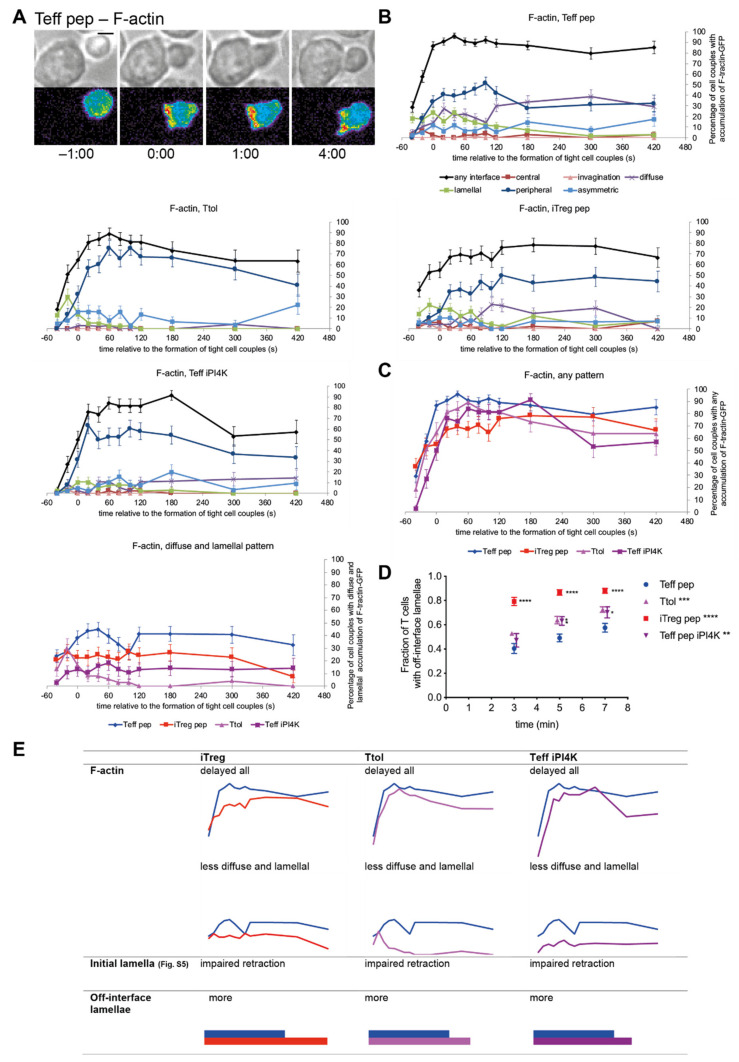
Actin-driven formation of a polarized cell couple is impaired upon induction of a regulatory phenotype. Tg4 T cells as indicated transduced to express F-tractin-GFP were activated with PL8 APCs (10 µg/mL Ac1-9[4Y]). (**A**) Representative F-tractin-GFP imaging data as in Figure 2A. A corresponding video is available as Video S7. (**B**,**C**) Data are expressed as the percentage of cell couples with F-tractin-GFP accumulation in the indicated patterns (Appendix A) relative to tight cell couple formation. Number of cell couples analyzed are n = 76, 38, 49, 38 for Teff pep, Ttol, iTreg pep, Teff 9608 cells, respectively, from 2–6 independent experiments. Statistical significance of differences between conditions is given in Appendix A; (**B**) All patterns for each cell type separately; (**C**) accumulation in any pattern (left) and in the diffuse and lamellal pattern (right) in direct comparison between the cell types. (**D**) The same data are expressed as the percentage of Tg4 T cells with off-interface lamellae, cumulative up to the indicated time relative to tight cell coupling. Some data points are slightly nudged to increase legibility. (**E**) Changes in accumulation phenotypes of the different Tg4 cell types with regulatory capability in comparison to Teff cells are summarized across all experiments with a cytoskeletal focus of the manuscript.

**Figure 6 cells-10-00418-f006:**
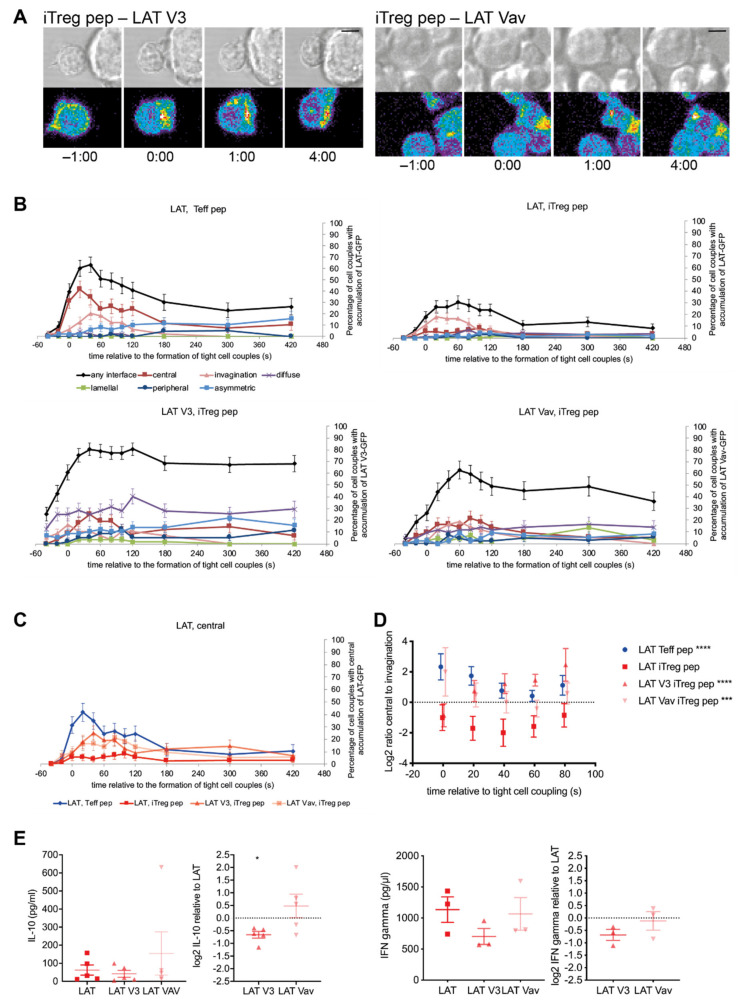
cSMAC formation can be partially restored with synthetic approaches in Tg4 iTreg pep cells. (**A**–**D**) Tg4 T cells as indicated transduced to express LAT-GFP, LAT V3-GFP or LAT-Vav were activated with PL8 APCs (10 µg/mL Ac1-9[4Y]). (**A**) Representative LAT V3-GFP and LAT-Vav imaging data as in Figure 2A. A video corresponding to the LAT-V3-GFP data is available as Video S8. (**B**,**C**) Data are expressed as the percentage of cell couples with LAT-GFP, LAT-V3-GFP or LAT Vav-GFP accumulation as noted in the indicated patterns (Appendix A) relative to tight cell couple formation. Number of cell couples analyzed are n = 57, 43 for iTreg pep cells expressing LAT V3-GFP and LAT Vav-GFP, respectively, from 4 and 2 independent experiments. LAT-GFP data are from Figure 2B. Statistical significance of differences between conditions is given in Appendix A; (**B**) all patterns for each cell type and LAT construct separately; (**C**) only central accumulation in direct comparison between the cell types and LAT constructs; (**D**) The same data are expressed as the log_2_ of the ratio of percent Tg4 T cells with accumulation in the central over the invagination pattern relative to tight cell couple formation. LAT-GFP data are from Figure 2D. Data points are slightly nudged to increase legibility. **E** Tg4 iTreg pep cells transduced to express LAT-GFP, LAT V3-GFP or LAT-Vav were activated with PL8 APCs (10 µg/mL Ac1-9[4Y]) for 18 h. Data are expressed as amount of IFNγ and IL-10 in tissue culture supernatants from 3 independent experiments, each, and also as the log2 of the ratio of IFNγ and IL-10 in iTreg pep cells expressing LAT V3-GFP or LAT Vav-GFP relative to those expressing LAT-GFP.

**Figure 7 cells-10-00418-f007:**
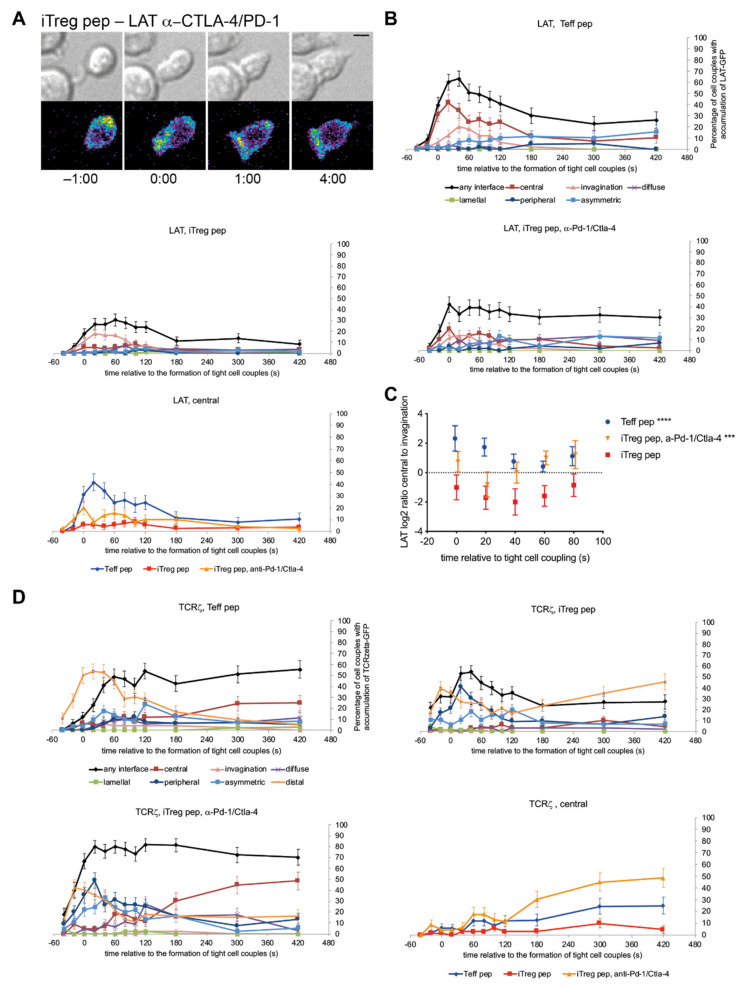
Pd-1 and Ctla-4 impair cSMAC formation upon induction of a regulatory phenotype. (**A**–**C**) Tg4 T cells as indicated transduced to express LAT-GFP were activated with PL8 APCs (10 µg/mL Ac1-9[4Y]). A Representative LAT-GFP imaging data upon treatment with 10 µg/mL α-Pd-1 plus α-Ctla-4 as in Figure 2A. A corresponding video is available as Video S9. (**B**) The data are expressed as the percentage of cell couples with LAT-GFP accumulation in the indicated patterns (Appendix A) relative to tight cell couple formation. Number of cell couples analyzed are n = 51 for iTreg pep cells in the presence of 10 µg/mL α-Pd-1 plus α-Ctla-4 from 3 independent experiments. Statistical significance of differences between conditions is given in Appendix A. Teff pep and iTreg pep data are from Figure 2B. The panel in the third row at the left is a direct comparison of central accumulation under the three experimental conditions. (**C**) The same data are expressed as the log_2_ of the ratio of percent Tg4 T cells with accumulation in the central over the invagination pattern relative to tight cell couple formation. Teff pep and iTreg pep data are from Figure 2D. Data points are slightly nudged to increase legibility. (**D**) Tg4 T cells as indicated transduced to express TCRζ-GFP were activated with PL8 APCs (10 µg/mL Ac1-9[4Y]). The graphs display the percentage of cell couples with TCRζ-GFP accumulation in the indicated patterns (Appendix A) relative to tight cell couple. Number of cell couples analyzed are n = 45 for iTreg pep cells in the presence of 10 µg/mL α-Pd-1 plus α-Ctla-4 from 2 independent experiments. Statistical significance of differences between conditions is given in Appendix A. Teff pep and iTreg pep data are from Figure 3B. The panel in the last row on the right is a direct comparison of central accumulation under the three experimental conditions.

## Data Availability

Imaging data are available at murphylab.cbd.cmu.edu/data/Treg2021.

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
