# Peer review of "A LAT-Based Signaling Complex in the Immunological Synapse as Determined with Live Cell Imaging Is Less Stable in T Cells with Regulatory Capability"

_cells, 2021, doi:10.3390/cells10020418_

Round 1
Reviewer 1 Report
The time course of the assembly and disintegration of a central supramolecular signaling complex (cSMAC) at the immunological synapse between antigen presenting cells and T cells was monitored by live cell imaging using GFP-tagged reporter molecules. For this purpose T cells with a T4RG T cell receptor (TCR), which recognizes the self-antigen myelin basic protein, were generated from transgenic mice using different protocols in order to obtain either effector T cells or three T cell types with regulatory capability.
1) This report contains a lot of data obtained in highly sophisticated experimental systems. Despite this wealth of descriptive findings, it is still not quite clear to me, what in particular the present comparative image analysis of immunological synapses involving T cell preparations with different regulatory capability will contribute to our understanding of immunology. Rather than a rather weak main conclusion, the title could specify the type of analysis and mention live cell imaging and immunological synapses.
2) Fig. 1 shows an outline of the preparation of effector and regulatory T cells and control experiments for their distinction. Given the complicated comparisons in this manuscript, the scheme in part A could be very helpful for readers. It should be restricted to the distinction of the investigated T cell preparations, while other methods necessary for the analysis of immunological synapses (e.g. transduction of GFP-tagged proteins or addition of antigen presenting cells) may be shown in the appropriate context. Labeling of the basic preparation scheme should be shown only once (i.e. NOT four times, namely for each preparation type). Only the distinctive features of the four protocols should be highlighted. In addition, I suggest short, illustrative and non-redundant designations for the four T cell types (e.g. Teff instead of Teff pep; Treg instead of iTreg pep; and PI4Ki instead of the drug name).
3) The control experiments in Fig. 1 should show more direct comparisons of the four preparations. For instance FoxP3 expression is shown as percentage of positive cells for effector and regulatory T cells and as mean fluorescence intensity for untreated and PI4K-treated effector T cells. This should be unified also for cytokine and surface protein expression. Some panels, e.g. B, D, G and J could be omitted or transferred to the supplement.
4) Fig. 2 shows the time course of cSMAC development in the four T cell types visualzed with GFP-tagged linker for activation of T cells (LAT). This is a sort of prototoype experiment that was also performed with other reporter molecules, namely a tandem C1 sensor for diacylglycerol (Fig. 3), F-actin (Fig. 4), LAT variants (Fig. 5). Finally the analysis was also performed in the presence of anti-PD1 and anti-CTLA4 antibodies. As it stands the resulting micrographs and time course plots look merely repetitive and it is extremely difficult for readers to extract any differences to the LAT results in Fig. 2. Therefore, the manuscript would profit from a scheme summarizing these experimental approaches. Importantly, a survey of the differences obtained with different reporter molecules and proximal signal modification should be provided to arrive at additional and more specific conclusions or findings warranting future research.
Author Response
We thank the reviewer for the helpful comments.
The time course of the assembly and disintegration of a central supramolecular signaling complex (cSMAC) at the immunological synapse between antigen presenting cells and T cells was monitored by live cell imaging using GFP-tagged reporter molecules. For this purpose T cells with a T4RG T cell receptor (TCR), which recognizes the self-antigen myelin basic protein, were generated from transgenic mice using different protocols in order to obtain either effector T cells or three T cell types with regulatory capability.
- This report contains a lot of data obtained in highly sophisticated experimental systems. Despite this wealth of descriptive findings, it is still not quite clear to me, what in particular the present comparative image analysis of immunological synapses involving T cell preparations with different regulatory capability will contribute to our understanding of immunology. Rather than a rather weak main conclusion, the title could specify the type of analysis and mention live cell imaging and immunological synapses.
We agree with the reviewer that the majority of our data are descriptive. However in our opinion, careful description of complex systems will enable future work, by others or us, to test the functional consequences of the described altered states of these complex systems. Here we hope to prompt such work. We have changed the title as suggested.
- 1 shows an outline of the preparation of effector and regulatory T cells and control experiments for their distinction. Given the complicated comparisons in this manuscript, the scheme in part A could be very helpful for readers. It should be restricted to the distinction of the investigated T cell preparations, while other methods necessary for the analysis of immunological synapses (e.g. transduction of GFP-tagged proteins or addition of antigen presenting cells) may be shown in the appropriate context. Labeling of the basic preparation scheme should be shown only once (i.e. NOT four times, namely for each preparation type). Only the distinctive features of the four protocols should be highlighted. In addition, I suggest short, illustrative and non-redundant designations for the four T cell types (e.g. Teff instead of Teff pep; Treg instead of iTreg pep; and PI4Ki instead of the drug name).
We have revised the scheme in Fig. 1A (and that in Fig. S2B) as suggested. We have changed the name for the UCB9608-treated T cells to Teff iPI4K. As we use effector and iTreg Tg4 T cells primed with peptide and antibody at multiple places across the manuscript, we believe that, to avoid confusion, denoting the method of priming consistently is useful despite being a bit cumbersome.
- The control experiments in Fig. 1 should show more direct comparisons of the four preparations. For instance FoxP3 expression is shown as percentage of positive cells for effector and regulatory T cells and as mean fluorescence intensity for untreated and PI4K-treated effector T cells. This should be unified also for cytokine and surface protein expression. Some panels, e.g. B, D, G and J could be omitted or transferred to the supplement.
We have moved representative data into the supplement as requested. In choosing how to quantify FACS data we take the nature of changes between experimental conditions into account. When two distinct populations are present and proportions between the populations shift, such as in the FoxP3 expression data in Fig. 1B, S1A, B or in the cytokine expression data in Fig. 1E, S1C, we display percentages of positive cells, as the MFI of the positive population doesn’t change significantly. However, when the expression of a protein shifts across the entire population, such as in FoxP3 expression upon treatment with UCB9608 in Fig. 1G, S1E, we display the MFI, as an arbitrary separation of the population into positive and negative cells, e.g. using the upper fluorescence limit of the negative control, to determine the percentage of positive cells seems less informative.
- 2 shows the time course of cSMAC development in the four T cell types visualized with GFP-tagged linker for activation of T cells (LAT). This is a sort of prototype experiment that was also performed with other reporter molecules, namely a tandem C1 sensor for diacylglycerol (Fig. 3), F-actin (Fig. 4), LAT variants (Fig. 5). Finally the analysis was also performed in the presence of anti-PD1 and anti-CTLA4 antibodies. As it stands the resulting micrographs and time course plots look merely repetitive and it is extremely difficult for readers to extract any differences to the LAT results in Fig. 2. Therefore, the manuscript would profit from a scheme summarizing these experimental approaches. Importantly, a survey of the differences obtained with different reporter molecules and proximal signal modification should be provided to arrive at additional and more specific conclusions or findings warranting future research.
Thank you for this very helpful suggestion. We have added a summary of all accumulation phenotypes under all experimental conditions as Figure 3C.
Reviewer 2 Report
In their study, Li et al. perform a detailed analysis of TCR proximal events in effector vs. tolerized T cells. To this end, they employ live cell imaging together with fluorescence based reporters. Of note, in all comparative experiments they employ the same TCR transgenic model, thus eliminating possible interclonal differences. They demonstrate that tolerized T cells display profound differences at the level of the cSMAC. Overall, this is a well-designed albeit somewhat descriptive study. However, the authors should work on their presentation of data to make their findings more accessible and more readily interpretable.
Criticism in detail:
Data presentation: Overall, graphical representations and font sizes are too small to be comfortably assessed. Current graphs preclude intuitive comparative assessment of different GFP accumulation patterns, because each individual graph represents the combination of patterns within one single cell type. Rather, to be able to directly compare the patterns between cell types, the latter should be combined in a single graph. This would also permit inclusion of statistical analysis in the main figures.
Figure 2: The authors conclude that LAT assembly is terminated faster in tolerized cells, but at least for Treg cells the downward slope appears to have a very similar steepness when compared to Teff.
Figure 3: What is the rationale to analyze DAG and TCRz in only two or three conditions, respectively, rather than in all four? The figure legend does not match the arrangement of individual panels. A) refers to the “bottom”, but it should be on the “right”.
Checkpoint inhibition: The authors should comment on the question, whether the observed effects are due to steric hindrance caused by the blocking antibodies or signal inhibition or both.
Methods: In 2.1 the authors indicate 6 steps of ten-fold dilutions, but the range of 0.08ug to 80ug requires only 4 steps.
Author Response
We thank the reviewer for the helpful comments.
In their study, Li et al. perform a detailed analysis of TCR proximal events in effector vs. tolerized T cells. To this end, they employ live cell imaging together with fluorescence based reporters. Of note, in all comparative experiments they employ the same TCR transgenic model, thus eliminating possible interclonal differences. They demonstrate that tolerized T cells display profound differences at the level of the cSMAC. Overall, this is a well-designed albeit somewhat descriptive study. However, the authors should work on their presentation of data to make their findings more accessible and more readily interpretable.
Criticism in detail:
Data presentation: Overall, graphical representations and font sizes are too small to be comfortably assessed. Current graphs preclude intuitive comparative assessment of different GFP accumulation patterns, because each individual graph represents the combination of patterns within one single cell type. Rather, to be able to directly compare the patterns between cell types, the latter should be combined in a single graph. This would also permit inclusion of statistical analysis in the main figures.
We have struggled how best to present patterning data for years. Grouping all patterns for one condition in a single panel makes comparisons between conditions more difficult, as pointed out by the reviewer. Grouping different conditions in a single panel forces us to omit the majority of patterns. To present all relevant data and allow straightforward comparisons across experimental conditions, we have now added a total of seven panels with direct comparisons of selected patterns to Figures 2, 4, 5 and 6.
Figure 2: The authors conclude that LAT assembly is terminated faster in tolerized cells, but at least for Treg cells the downward slope appears to have a very similar steepness when compared to Teff.
There are two possible measures of how quickly central LAT accumulation is terminated, the downward slope of central patterning after peak accumulation (Figure 2C) and the ratio of central to invagination accumulation (Fig. 2D). We would argue that the peak of central LAT accumulation in iTreg cells (as opposed to the Ttol and Teff iPI4K) is so small that it is difficult to determine a meaningful downward slope. Therefore, we rely on the ratio of central to invagination accumulation that in the iTregs is the smallest of all conditions over the first 40 seconds after the time of tight cell coupling. We now note explicitely that a rapid transition into the invagination is indicative of ill sustained LAT accumulation in a cluster of supramolecular signaling complexes at the interface center.
Figure 3: What is the rationale to analyze DAG and TCRz in only two or three conditions, respectively, rather than in all four? The figure legend does not match the arrangement of individual panels. A) refers to the “bottom”, but it should be on the “right”.
We apologize for the incomplete nature of the data presented in Figure 3. There is no good rationale. Some of the data included in the figure have been acquired years ago and for practical reasons it was very difficult to complete the data set more recently. We have corrected the figure legend.
Checkpoint inhibition: The authors should comment on the question, whether the observed effects are due to steric hindrance caused by the blocking antibodies or signal inhibition or both.
This question is now address as follows: “The effect of the antibodies is likely caused by the direct blockade of ligand binding of Ctla-4 and Pd-1 by the antibodies rather than by steric interference with synapse organization by the antibody/inhibitory receptor complexes, as dextran with a diameter of 13 nm, larger than an antibody, can readily move through the interface between a Jurkat T cell with a Raji APC [45].”
Methods: In 2.1 the authors indicate 6 steps of ten-fold dilutions, but the range of 0.08ug to 80ug requires only 4 steps.
We apologize for the lack of clarity. The section has been revised to read: “in a dose escalation regime from 0.08 µg to 80 µg peptide in 4 treatments of 10-fold increases in peptide amount per treatment followed by two more treatments at the highest dose.”
Round 2
Reviewer 1 Report
The time course of the assembly and disintegration of a central supramolecular signaling complex (cSMAC) at the immunological synapse between antigen presenting cells and T cells was monitored by live cell imaging using GFP-tagged reporter molecules. For this purpose T cells with a T4RG T cell receptor (TCR), which recognizes the self-antigen myelin basic protein, were generated from transgenic mice using different protocols in order to obtain either effector T cells or three T cell subtypes with regulatory capability. In addition, the roles of LAT, DAG, F-actin and checkpoint inhibitors were interrogated by using different labelled reporter molecules and CTLA4 and PD1 antibodies.
1) Revision resulted in an improved title of the manuscript, but the manuscript remains a collection of preliminary, incompletely evaluated observations rather than a study with definitive descriptive results and clear conclusions.
2) Some of the excessive labelling in Fig. 1A was removed, but this scheme is still redundantly labelled and contains information that belongs to the Methods section.
3) Unifying measures for comparisons are missing overall. This starts in the control experiments of Fig. 1, where different measures for FOXP3 expression are used for different T cell preparations.
4) Some Figure parts were transferred to the supplement, but this process is still incomplete. For example the differences between transduced and non-transduced T cell preparation (Fig. 1E) are merely technical control experiments that do not contribute to the presentation of the project and therefore also must be moved to the supplement. A selection of the data that are necessary to present for adequate description of cSMAC development at the immunological synapse still needs to be made.
5) Some comparisons between the different T cell preparations have been introduced, but a comprehensive, quantitative assessment of the differences in cSMAC assembly and disintegration under different conditions is still missing. The vague statement of reduced cSMAC stability is not sufficiently substantiated and differences in the achieved accumulation of signaling molecules are ignored.
6) Also after the revision, data are repetitively presented as micrographs and time course plots for separate T cell preparations. It would be more constructive to show, how the T cell preparations differ in selected parameters. Since the time course of cSMAC development in the four T cell types was visualized with GFP-tagged linker for activation of T cells (LAT) as a sort of prototoype experiment and repeated with other reporter molecules, i.e. a tandem C1 sensor for diacylglycerol (Fig. 3), F-actin (Fig. 4) and LAT variants (Fig. 5), the manuscript would also profit from a scheme summarizing the roles of different reporter molecules and proximal signal modification.
7) A summarizing Table such as inserted as Fig. 3 C could indeed be helpful to define a direction of the manuscript. Since it summarizes the whole manuscript, it should, however, be inserted as a separate item after the mentioned Figures. Instead of vague explanatory text, this Table must contain quantitative Figures for comparing effector T cells with those with regulatory capabilities.
8) In summary, I found that the revision was starting in a desirable direction, but by no means was thorough enough. Although this manuscript provides a wealth of preliminary, only partly evaluated data in sophisticated molecularly defined systems, it is practically impossible for readers who are not directly involved in the project to appreciate these descriptive findings. It is still necessary to select a consistent way of data selection and presentation to arrive at sizable conclusions to advance the field.
Author Response
We again thank the reviewer for the helpful comments.
1) Revision resulted in an improved title of the manuscript, but the manuscript remains a collection of preliminary, incompletely evaluated observations rather than a study with definitive descriptive results and clear conclusions.
We agree with the reviewer that the manuscript consists of a large set of comparative observations. As argued in our response to the first set of review comments, careful description of complex systems will enable future work, by others or us, to test the functional consequences of the described altered states of these complex systems. We respectfully disagree that the observations are preliminary and incompletely evaluated. The manuscript is based on the investigation of 1317 single cell movies of activating T cells, an analysis of more than 15,000 T cell:APC images for protein distributions in 3D. To us this is a more than preliminary scope. We have quantified the imaging data with two different approaches that both are extensively validated by previous publications. Statistically significant differences consistently emerge. We fail to see where this remains incomplete. We appreciate that the description of complex systems more often than not doesn’t yield simple pictures. In the manuscript we try to balance a detailed view at the comprehensive imaging data as a community resource with numerous figure panels (Figs. 2C, 2D, 4, 5C, 5D, 5E, 6C, 6D, 7B, 7C, 7D, S8B) to focus on key themes therein. We are grateful for the suggestion to make common themes even more accessible, as further discussed below. However, we believe that sacrificing system complexity for the sake of clear conclusions isn’t necessarily in the best interest of science.
2) Some of the excessive labelling in Fig. 1A was removed, but this scheme is still redundantly labelled and contains information that belongs to the Methods section.
We have replacd 50 U/ml and 100 U/ml IL-2 with IL-2 and IL-2hi.
3) Unifying measures for comparisons are missing overall. This starts in the control experiments of Fig. 1, where different measures for FOXP3 expression are used for different T cell preparations.
We have argued in response to the first set of review comments that different measures of FoxP3 expression are most appropriate for the different types of experiments. To repeat, when two distinct populations are present and proportions between the populations shift, such as in the FoxP3 expression data in Fig. 1B, S1A, B or in the cytokine expression data in Fig. 1E, S1C, we display percentages of positive cells, as the MFI of the positive population doesn’t change significantly. However, when the expression of a protein shifts across the entire population, such as in FoxP3 expression upon treatment with UCB9608 in Fig. 1G, S1F, we display the MFI, as an arbitrary separation of the population into positive and negative cells, e.g. using the upper fluorescence limit of the negative control, to determine the percentage of positive cells seems less informative. We are not sure why this argument is insufficient. The measures to quantify the imaging data are entirely uniform across the manuscript. We are not sure how this constitutes that unifying measures for comparison are missing overall.
4) Some Figure parts were transferred to the supplement, but this process is still incomplete. For example the differences between transduced and non-transduced T cell preparation (Fig. 1E) are merely technical control experiments that do not contribute to the presentation of the project and therefore also must be moved to the supplement. A selection of the data that are necessary to present for adequate description of cSMAC development at the immunological synapse still needs to be made.
We have split Fig. 1E to display the GFP-positive data only in the main figure and moved the GFP-negative data into a supplement alongside a repetition of the GFP-positive data. As argued above, in the manuscript we try to balance a detailed view at the comprehensive imaging data as a community resource with numerous figure panels (Figs. 2C, 2D, 4, 5C, 5D, 5E, 6C, 6D, 7B, 7C, 7D, S8B) to focus on key themes therein. To us this represents a selection of key data.
5) Some comparisons between the different T cell preparations have been introduced, but a comprehensive, quantitative assessment of the differences in cSMAC assembly and disintegration under different conditions is still missing. The vague statement of reduced cSMAC stability is not sufficiently substantiated and differences in the achieved accumulation of signaling molecules are ignored.
We have consistently documented more transient and sometimes diminished accumulation of LAT as a critical cSMAC component in three different ways (as illustrated here with the respective panels of Fig. 2). We display the overall pattern analysis for each condition with an extensive statistical analysis in supplementary tables (Fig. 2B). In our first revision, we have now introduced direct comparisons of central accumulation data and are grateful for this suggestion (Fig. 2C). We use the ratio of central to invagination accumulation to further characterize the labile nature of LAT accumulation in T cells with regulatory capability (Fig. 2D). We are not sure how this constitutes a vague and insufficient characterization. We agree with the reviewer that these are key changes in the context of additional ones. We are uncertain how to combine the earlier request for a more focused story with the request to consider additional changes here. We believe that our balance between access to full data complexity and emphasis of key themes is about right.
6) Also after the revision, data are repetitively presented as micrographs and time course plots for separate T cell preparations. It would be more constructive to show, how the T cell preparations differ in selected parameters. Since the time course of cSMAC development in the four T cell types was visualized with GFP-tagged linker for activation of T cells (LAT) as a sort of prototoype experiment and repeated with other reporter molecules, i.e. a tandem C1 sensor for diacylglycerol (Fig. 3), F-actin (Fig. 4) and LAT variants (Fig. 5), the manuscript would also profit from a scheme summarizing the roles of different reporter molecules and proximal signal modification.
7) A summarizing Table such as inserted as Fig. 3 C could indeed be helpful to define a direction of the manuscript. Since it summarizes the whole manuscript, it should, however, be inserted as a separate item after the mentioned Figures. Instead of vague explanatory text, this Table must contain quantitative Figures for comparing effector T cells with those with regulatory capabilities.
We are grateful for the suggestion to make our short summary figure panel more comprehensive and more visually accessible. We have now generated a separate summary figure (Figure 4) for all data relating to cSMAC stability as underpinned by outline summary graphs. In addition, we have added a similar figure for the cytoskeletal phenotypes as Fig. 5E.
8) In summary, I found that the revision was starting in a desirable direction, but by no means was thorough enough. Although this manuscript provides a wealth of preliminary, only partly evaluated data in sophisticated molecularly defined systems, it is practically impossible for readers who are not directly involved in the project to appreciate these descriptive findings. It is still necessary to select a consistent way of data selection and presentation to arrive at sizable conclusions to advance the field.
We are grateful that the reviewer sees our revision moving in the right direction and have further revised the manuscript along the lines detailed above. We are afraid that increasing complexity of data, with all the necessary effort to make them accessible as greatly aided by the review suggestions, often leave the reader with stories requiring a bit more effort to appreciate. More comprehensive understanding may well make this worthwhile.